# Laser Bioprinting with Cell Spheroids: Accurate and Gentle

**DOI:** 10.3390/mi14061152

**Published:** 2023-05-30

**Authors:** Ekaterina D. Minaeva, Artem A. Antoshin, Nastasia V. Kosheleva, Polina I. Koteneva, Sergey A. Gonchukov, Svetlana I. Tsypina, Vladimir I. Yusupov, Peter S. Timashev, Nikita V. Minaev

**Affiliations:** 1Institute of Photon Technologies of FSRC «Crystallography and Photonics» RAS, Troitsk, 108840 Moscow, Russia; minaeva.e.d@bk.ru (E.D.M.); antoshin_a_a@staff.sechenov.ru (A.A.A.); tsypina@yandex.ru (S.I.T.); iouss@yandex.ru (V.I.Y.); 2National Research Nuclear University MEPhI (Moscow Engineering Physics Institute), 115409 Moscow, Russia; sagonchukov@mephi.ru; 3World-Class Research Center “Digital Biodesign and Personalized Healthcare”, Sechenov University, 8-2 Trubetskaya St., 119991 Moscow, Russia; timashev_p_s@staff.sechenov.ru; 4Institute for Regenerative Medicine, Sechenov University, 8-2 Trubetskaya St., 119991 Moscow, Russia; kosheleva_n_v@staff.sechenov.ru (N.V.K.); koteneva_p_i@staff.sechenov.ru (P.I.K.); 5FSBSI Institute of General Pathology and Pathophysiology, 8 Baltiyskaya, 125315 Moscow, Russia

**Keywords:** bioprinter, laser induced forward transfer, laser assisted bioprinting, cell spheroids, Pi-shaper

## Abstract

Laser printing with cell spheroids can become a promising approach in tissue engineering and regenerative medicine. However, the use of standard laser bioprinters for this purpose is not optimal as they are optimized for transferring smaller objects, such as cells and microorganisms. The use of standard laser systems and protocols for the transfer of cell spheroids leads either to their destruction or to a significant deterioration in the quality of bioprinting. The possibilities of cell spheroids printing by laser-induced forward transfer in a gentle mode, which ensures good cell survival ~80% without damage and burns, were demonstrated. The proposed method showed a high spatial resolution of laser printing of cell spheroid geometric structures at the level of 62 ± 33 µm, which is significantly less than the size of the cell spheroid itself. The experiments were performed on a laboratory laser bioprinter with a sterile zone, which was supplemented with a new optical part based on the Pi-Shaper element, which allows for forming laser spots with different non-Gaussian intensity distributions. It is shown that laser spots with an intensity distribution profile of the “Two rings” type (close to Π-shaped) and a size comparable to a spheroid are optimal. To select the operating parameters of laser exposure, spheroid phantoms made of a photocurable resin and spheroids made from human umbilical cord mesenchymal stromal cells were used.

## 1. Introduction

Currently, tissue engineering is a promising and popular approach to the restoration of damaged tissues and organs, as opposed to the traditional approach to transplantation of donor material, where demand is much higher than supply, and the transplanted material can cause immune responses [1]. For the purposes of tissue engineering, autologous cells are most often used both in the form of standard suspension forms and in the form of aggregates (spheroids). Cell spheroids are 3D spherical aggregates of cells and have a number of advantages over cell suspensions and 2D cultures: high density, similar to that of native tissues [2], preformed cell contacts and microenvironment, as well as ease of manipulation, which enables their use as building blocks [3] for the production of tissues and organs.

Bioprinting is one of the perspective methods for making tissues and organs from spheroids [4,5]. The most widely used bioprinting techniques, inkjet and extrusion, are nozzle-based, and they are the cheapest and most available in the market. However, the nozzle can significantly limit the maximum printing speed and potential bioink viscosity range, while the created shear stress was reported to decrease post-printing spheroid viability [6]. Moreover, these approaches do not provide sufficient printing resolution, which is essential for building complex organs and tissues [6].

As well, the nozzle-free high-precision techniques, such as aspiration [7,8], acoustic [9], and Kenzan-based [10] bioprinting, have been reported for spheroid-based biofabrication. Despite having strong advantages, they still possess some limitations. For example, aspiration bioprinting requires physical movement of the suction nozzle from the reservoir with spheroids to the area with the printed construct, which takes about 20 s for each cycle, and in this case, the process of a large spheroid-based construct can take about 2 days [8]. For acoustic bioprinting [9], there is also a need to apply each individual spheroid to the transfer device (focused interdigital transducers) using manual or automated transfer. In both variants, the bioprinting will be quite a long process. Finally, the Kenzan-bioprinting method is limited by usable spheroid size, inability to print complex shaped structures, and inability to work with individual cells [7,10].

Laser-induced forward transfer (LIFT) [11,12,13,14] is another bioprinting technique that can surpass the above-mentioned approaches and overcome their drawbacks. Applying cell suspensions, it has shown the ability to print the constructs with micron-scale precision and at high speed from the bioinks with a wide viscosity range [15] while supporting sufficient post-printing cell viability, which makes this technique a candidate for spheroid bioprinting [16,17,18,19]. However, LIFT has not been applied for this purpose yet. Of note, it has been previously used to transfer the spheroid-like structures (“microbeads”) by [20], but these microbeads were formed by gelled alginate and, therefore, more resistant to mechanical stress rather than “real” spheroids used for creating tissues and organs [3]. Therefore, bioprinting with cell spheroids using the LIFT technique still remains an unsolved challenge.

For delicate transfer of a spheroid using a LIFT without significant damage, it is necessary to: (1) ensure minimum radial pressure gradients; (2) minimize heating of the spheroid during bioprinting; (3) minimize the kinetic energy of transfer in order to prevent impact damage; (4) provide a good resolution of the method and a high packing density of spheroids. However, it is practically impossible to simultaneously satisfy these conditions using standard laser bioprinters, since they traditionally use laser beams with a Gaussian energy distribution profile in the spot. The use of Gaussian beams with a small waist diameter can lead to significant damage to the spheroid during printing due to the high fluence at the center of the laser spot. At the same time, an increase in the waist radius of the laser beam will lead to an increase in the transferred hydrogel droplet. This will significantly degrade the resolution of the bioprinting method, as well as increase the consumption of the gel surrounding the spheroids, reducing the overall resolution of the printing process [21].

In the study by Liang et al. [22], a beam-shaping laser-induced forward transfer (BS-LIFT) technique was presented, which makes it possible to form a complex-shaped laser spot for the transfer of multicellular objects with a sophisticated structure. The approach allows for forming a sophisticated structure from several small laser spots, which are distributed over the area of a transported multicellular object and specified by the mathematical phase pattern of the spatial light modulator. Such a solution is beneficial for the transfer of 2D extended cell aggregates, but it is difficult to apply to 3D cell aggregates (cell spheroids).

In our opinion, the gentle mode of large three-dimensional object transfer, such as cell spheroids, can be implemented by using a complex, non-Gaussian laser beam shape with a tunable smooth energy distribution over the spheroid cross-section. In this paper, we have demonstrated the potential of this approach to provide high resolution and survivability in laser printing with spheroids.

## 2. Materials and Methods

### 2.1. Experimental Setup for Bioprinting with Cell Spheroids

The scheme of the experimental setup for LIFT of cell spheroids is shown in Figure 1.

The experiments were performed using a pulsed nanosecond diode-pumped solid-state Q-switched laser TECH-1053 (Laser-compact Plus Co., Ltd., Moscow, Russia) with a wavelength *λ* = 1053 nm, pulse duration *τ* < 7 ns, beam quality *M^2^* < 1.2 and pulse energy *E* < 0.8 μJ. The nir-IR wavelength was chosen to reduce the negative effects on cells. The low energy of photons with λ = 1053 nm and the relatively long duration of laser pulses (τ < 7 ns), which do not provide multiphoton absorption, do not lead to the direct destruction of chemical bonds [23]. The studies carried out in [24] showed that even the use of shorter wavelength ultraviolet radiation of 355 nm with a pulse duration of 10 ns (approximately as in our work) leads to negligible genotoxic stress during laser bioprinting. Two optical elements—a telescopic lens with a magnification range of 2×–8× and a motorized beam expander with a magnification range of 1×–3× (Optogama, Vilnius, Lithuania) were used to adjust the beam diameter and change its divergence. A motorized beam expander allows us to control the size and divergence of the laser beam from a computer.

An optical element, “Focal-piShaper-9-1064HP” (AdlOptica Optical Systems GmbH, Berlin, Germany), was used to form non-Gaussian intensity distributions in the laser spot. Thus, the formation of intensity distribution profiles Π-shaped such as “donut”, “small flat-top”, “flat-top”, or a set of concentric rings with different diameters is possible using Pi-Shaper. Manual adjustment of the Pi-Shaper in combination with the settings for the divergence and magnification of the laser beam diameter on the motorized beam expander allows for obtaining the required types of the intensity distribution. The required intensity distribution profile was formed in the focal plane of the F-theta lens with a diameter close to that of a cell spheroid.

It is necessary to set the spot size and the shape of the intensity distribution in the laser spot to adjust the system for the required laser bioprinting mode. For this, one should sequentially adjust the laser beam diameter at the entrance to the Pi-Shaper and the divergence of the laser irradiation, as well as the distance of the working plane (absorbing layer on the donor plate) from the focal point. According to Laskin et al. [25], non-Gaussian distribution patterns are formed not in the Pi-Shaper focal plane itself but before and after it. We used a laser spot formed before the focal plane because the influence of the laser irradiation divergence on the intensity distribution is smaller than in the region after the focal plane. The distance between different intensity profiles in the laser spot [25] in the real case was small and ranged from 0.57 to 1.71 mm. Therefore, tuning to the required intensity distribution (changing the position of the focal plane) was carried out by changing the laser beam divergence on a motorized expander without moving the donor plate with spheroids.

The following things should be taken into account in order to obtain the required intensity distribution in the laser spot. Increasing the diameter of the laser spot at the entrance to the Pi-Shaper allows the transition from a Gaussian distribution to a pseudo “Π-shaped” profile with intensity modulation at its top [25]. The intensity distribution in the laser spot will have a strong modulation with minima and maxima when using a laser spot close to the maximum diameter at the Pi-Shaper input. Such tuning can cause excessive pressure gradients in laser bioprinting. To carry out the experiments, it was necessary to choose the diameter of the laser beam at the Pi-Shaper entrance, under which two conditions were met for the formed spot on the absorbing layer of the donor substrate: (1) the diameter of this spot should approximately correspond to the diameter of the transferred spheroid, and (2) the intensity distribution profile of the laser radiation should be roughly Π-shaped with moderate contrast at the top. The intensity distributions in the laser spot used in the experiments were obtained with input beam diameters from 4.5 to 5 mm on the Pi-Shaper input aperture.

Two types of non-Gaussian laser irradiation intensity distribution in the spot were used in the experiments—“Ring and Dot” (RD) and “Two Rings” (TR) modes (Figure 1). 

The laser beam was focused using an SLF-1064-100-8 F-theta objective (Ronar-Smith, Singapore, Singapore) with a focal length of 100 mm. A single-mirror galvanic scanner LScanXY (Ateko-TM, Moscow, Russia) was used for the movement of the laser spot along the XY plane.

A Laser bioprinting setup [21,26,27], with similar functionality and optical design (F-theta lens and galvanoscanner) but without additional optical elements, was used for experiments with Gaussian intensity distribution in the laser spot. The radiation of the laser source YLPM-1-4x200-20-20 (NTO IRE-Polus, Fryazino, Russia) with a wavelength *λ* = 1064 nm, beam quality *M^2^* < 1.2, pulse duration *τ* = 8 ns and pulse energy *E* < 150 μJ was used in this setup.

The laser pulse energy was measured with a power meter (PM100D, Thorlabs, Newton, NJ, USA) with a pyroelectric sensor (ES120C, Thorlabs). BeamOn WSR sensor (Duma Optronics, Nesher, Israel) with a pixel size of 8 × 8 μm was used to measure the diameters of the output laser beam.

The processes occurring on the donor plate under laser irradiation were observed using a Toupcam XFCAM1080PHD digital camera (Touptec, Hangzhou, China). An additional long-focus macro lens made it possible to obtain an image from a 4 × 6 mm area. The camera was at an angle of 60° to the laser radiation propagation axis, which caused the spherical objects on the donor plate to look elliptical in photographs and videos. For the high-speed optical recording of the process of laser-induced formation of a jet of the culture medium and the transfer of a cell spheroid, a Fastcam SA-3 high-speed camera (Photron, Tokyo, Japan) was used. The recording was carried out in shadow photography mode with front illumination and a framerate of 10,000 fps.

### 2.2. Sterile Zone for Cell Spheroids Laser Bioprinting

The laser bioprinting process was organized inside an isolated area under sterile conditions (Figure 2) assembled from two Petri dishes—a lid diameter of 90 mm (1) and a bottom diameter of 60 mm (2). This construction allows moving the lid relative to the bottom in a horizontal plane and maintaining sterile conditions inside the Petri dish. Such a movement was necessary to implement the movement of the donor plate with spheroids relative to the acceptor plate without breaking the sterility conditions. A layer of sterile medical vaseline (3) was applied to the junction of the Petri dishes, which ensured the tightness of the internal volume.

Glass plates 1 mm thick were used as a donor plate (4). An absorbing titanium metal layer (5) was deposited on the donor plastic by magnetron sputtering with a thickness of 80 nm. Microscopic coverslips 20 × 20 mm in size were used as an acceptor plate (6), on which a layer of receiving gel (8) with a volume of 100 μL was applied. Cell spheroids (9) were placed into 25 μL of bioink (7), and it was applied over the absorbing layer of the donor plate, and then placed into the glass washer. The presence of a gasket in the form of a glass washer (10) with a thickness of 170 μm and an inner diameter of 10 mm allowed for: (1) forming a bioink layer of the required thickness and (2) limiting the spreading of the bioink over the surface of the donor plate. Wetted cotton swabs were placed into the volume of the Petri dish to prevent the hydrogel from drying out (11).

The donor plate was attached to the lid of the Petri dish using magnets. Flat magnets (12) were attached to the non-working surface of the donor plate with glue. The acceptor plate was placed on the bottom of the Petri dish. The distance between the donor and acceptor plates was set at a distance of 0.5–0.8 mm.

A motorized XY table based on MGN7C (HIWIN) rail guides and stepper motors with a screw shaft with a thread pitch of 0.5 mm was used to move the donor and acceptor substrates fixed on the bottom and lid of the Petri dish in the horizontal plane. The bottom and lid of the Petri dish were attached to the moving platforms, which allowed for moving the donor and acceptor plates relative to the focusing point of laser radiation without disturbing the microclimate inside and maintaining the distance between the donor and acceptor plates. The system was controlled by two BigTreeTech SKR 1.3 Pro controllers. The capabilities of the control system allow one to move plates both manually using joysticks and using a control program based on G-codes. The relative positioning accuracy of the samples was no less than 5 μm.

### 2.3. Laser Exposure Modes Setting and Characterization 

A preliminary selection of energy parameters and intensity distribution profiles in the laser spot was performed using cell spheroid phantoms. Cell spheroid phantoms were made from a photocurable 3D-printing polymer (F39T Resion, Dongguan, China) with a diameter of 150 μm. A high-speed video camera was used to register the phantom transfer process. The phantom transfer process was recorded using a high-speed camera, and then the transfer mode was determined according to visualized jet and the laser exposure parameters [21]. The mode of laser exposure was selected to ensure stable transfer of the spheroid phantom, after which the energy of the laser pulse was increased by 20% (the value was selected based on a series of test experiments on spheroid phantom and cell spheroids). The obtained value was set as the operating mode of laser printing by cellular spheroids. At the same time, a further increase in energy was considered inexpedient since it could lead to unstable transport modes and worsen the survival of spheroids. The result of exposure to the spheroid in terms of its damage from the laser pulse was evaluated using a digital camera with a microlens directly on the laser bioprinting unit, as well as using an HRM-300 microscope (Huvitz, Anyang, Republic of Korea) with differential interference contrast mode. 

Several types of laser spots with both Gaussian and non-Gaussian intensity distributions were used to refine the laser transfer technology (Table 1).

In the first case, a laser spot with a Gaussian energy distribution of small size with a diameter *D_TG_* = 30 μm (Thin Gauss) was used. Such a laser spot is usually used for the bioprinting of individual cells [12,13,28]. In the second case, a laser spot with a Gaussian energy distribution was also used, while its diameter was close to the size of a cell spheroid *D_AG_*~150 μm (Adapted Gauss). In the third case, a laser spot with a Gaussian energy distribution of large diameter *D_WG_*~1 mm (Wide Gauss) was used.

We were guided by the following hypotheses for choosing the parameters for conducting experiments on laser bioprinting. For gentle transfer of cell spheroids using a non-Gaussian energy distribution profile in a laser spot, it is necessary:To set the size of the laser spot close to the size of the diameter of the cell spheroid;To minimize the pressure gradients in the cell spheroid area by redistributing the energy maxima of the laser impact on the peripheral region of the spheroid;To reduce the energy of laser exposure in the area where the spheroid is located closest to the absorbing layer on the donor plate (in the center of the laser spot) to reduce the negative impact of high pulsed temperatures near the absorbing layer on the cell spheroid [29].

Two types of the non-Gaussian distribution of laser energy in the spot were proposed to implement these requirements, which allowed the formation of the Pi-Shaper optical element (Table 1): “Ring and Dot” mode and “Two Rings” mode.

The laser spot intensity distribution on the donor substrate was measured using the profile meter (Figure 3). 

### 2.4. Methodology for Calculating Laser Irradiation for Non-Gauss Intensity Distributions 

The non-Gaussian intensity distribution in the laser spot, for which the standard methods for calculating the fluence are not applicable, was used in the experiments. Therefore, we proposed a technique based on the calculation of the fluence along the axial (XY) sections of the laser radiation intensity distributions obtained using the Laser Beam Profiler. Evaluations were made based on the ellipticity approximation of the spots (Figure 3). It was assumed that the XY axes are directed along the minor and major axes of the ellipses.

The following algorithm was used for calculations: the dependences of the intensity on the radius with the beginning at the center of the laser spot for each of the axes were used as initial data, from which the averaged dependence of the intensity distribution on the radius for each laser spot was obtained. The fluence was calculated over the entire spot using the obtained average dependence in the condition of radial symmetry, which made it possible to obtain the fluence distribution over the spot. Based on these data, with a successive increase in the radius *i* ≥ 1 (in pixels), the average fluence $\bar{F_{au}}$ (in relative units) was determined inside the circular region of the laser spot for each radius *i*:$$\bar{F_{au}}\left(i\right)=\frac{I{\left(i-1\right)}^{2}+2iI\left(i\right)}{i^{2}}$$
where *I*—the intensity of laser radiation, *i*—the radius in pixels measured from the center of the spot. The laser spot diameter *D* was determined by the point where the $\bar{F}\left(i\right)$ dependence curve had the sharpest decrease with increasing radius *i.* The fluence was calculated in absolute units (J/cm^2^) under the assumption that the entire energy of the laser pulse *E* fell into a spot with a diameter *D*: $F=\frac{4E}{\pi D^{2}}$.

The proposed method for calculating the fluence was tested for a Gaussian spot with parameters used in the classical implementation of LIFT for printing with single cells *E* = 30 μJ, *D* = 30 μm. As a result, the average fluence calculated by the method described above differed from the value calculated by the “classic” formulas by only 3%.

### 2.5. Bioink Preparation

Bioprinting was performed using human umbilical cord-derived mesenchymal stromal cells (MSCs)-based spheroids. MSCs primary cultures were obtained from umbilical cord samples as described in the patent [30] with the approval of the Local Ethical Committee of Sechenov University. Obtained MSCs were cultured in Dulbecco’s Modified Eagle’s Medium (DMEM)/F12 (1:1, Biolot, St. Petersburg, Russia) supplemented with 10% fetal calf serum (HyClone, Wilmington, NC, USA) insulin–transferrin–sodium selenite (1:100, Biolot, Saint-Petersburg, Russia), bFGF (20 ng/mL, ProSpec, Ness-Ziona, Israel), heparine 5000 IU/mL (Belmedpreparaty, Minsk, Belarus), and gentamicin (50 g/mL, Paneco, Moscow, Russia) at 37 °C, 5% CO_2_. Cell morphology and viability were routinely checked with a phase-contrast microscope Axio Vert A1 (Carl Zeiss, München, Germany). Cell spheroids (2000 cells per spheroid) were formed as thoroughly described previously [30]. Briefly, cell suspension (3.4 × 10^6^ cells/mL) was placed in agarose non-adhesive microplates created with 3D Petri Dish molds (Microtissues, Providence, RI, USA), left at 37 °C, 5% CO_2_ for setting, and then covered with full growth medium in 12-well culture plates. Spheroids were cultured for 7 days at 37 °C, 5% CO_2_, before bioprinting.

For bioink preparation, high molecular weight hyaluronic acid (HA) powder *M_w_* = 1.6–1.8 MDa (Contipro, Dolní Dobrouč, Czech Republic) was sterilized at 121 °C for 15 min, diluted with DMEM/F12 cell medium (Biolot, Saint-Petersburg, Russia), and mixed with spheroids. The final HA concentration in bioinks was 2 mg/mL, and the spheroid concentration was 1600 units/mL.

To form the hydrogel layer on the acceptor plate, the fibrinogen (Sigma-Aldrich, St. Louis, MO, USA) was conjugated with PEG (Sigma-Aldrich, St. Louis, MO, USA) at a mass ratio of 1:0.06 in PBS, and their mixture was subsequently cultivated for 2 h at 37 °C. Next, the HA was added to this mixture so that the final concentration of the components was as following: PEG-fibrinogen—25 mg/mL, HA—5.6 mg/mL. Post printing, the acceptor plate solidification was achieved via thrombin solution (33 U/mL) spraying (Sigma-Aldrich, St. Louis, MO, USA).

#### Spheroid Viability Analysis

As a control, cell spheroids remaining on the donor plate were pipetted on an acceptor plate using a micropipette. The bioprinted spheroids in the solidified gel were further covered with full growth medium and incubated for 3 days at 37 °C, 5% CO_2_. Further, the live/dead assay was performed to visualize the viability of cells in spheroids. For this, they were stained at 37 °C in 5% CO_2_ for 30 min. Live cells stained with Calcein-AM (Sigma-Aldrich, St. Louis, MO, USA), dead ones stained with Propidium Iodide (Thermofisher, Waltham, MA, USA), while Hoechst 33258 (Thermofisher, Waltham, MA, USA) stained nuclei. Spheroids’ visualization was performed on a confocal laser scanning microscope LSM 800 (Carl Zeiss, Oberkochen, Germany).

## 3. Results and Discussion

### 3.1. Features of Laser Transfer of Cellular Spheroids and Their Phantoms Using the Gaussian Intensity Distribution Profile

In the first stage, a preliminary selection of laser exposure parameters was carried out using five options for the distribution of laser radiation intensity in the spot (Table 1) with Gaussian and non-Gaussian profiles. Such a selection was carried out by transferring cell spheroids and their polymer phantoms.

For the first variant, “Thin Gauss” (TG) (Table 1), which corresponds to the “standard” parameters for printing with single cells, it was found that for vertical transfer along the laser beam axis, it is necessary to act strictly at the center of the spheroid. Insignificant deviations from the center of the spheroid (~5 μm) lead to its drift relative to the axis of the laser beam, with a significant deterioration in the resolution of laser bioprinting. It was difficult to predict and correctly adjust the flight trajectory of both phantoms and the cell spheroids, so the “standard” version of laser exposure was considered unsuitable. For the transfer of cell spheroids, the first variant with a narrow laser spot turned out to be unsuitable also because such an action led to the destruction of the cell spheroid structure.

The second variant, “Adapted Gauss” (AG), with a spot diameter approximately corresponding to the diameter of the transported objects (Table 1), was suitable for both the transfer of phantoms and the transfer of cell spheroids according to visual data from the high-speed video recording. However, according to visual observations of cell spheroids after the transfer process, areas of thermal damage and, in some cases, ruptures of the cell spheroid were observed. This variant, which provides a relatively stable transfer of spheroids, was subsequently chosen as one of the options for the study of the survival of cell spheroids.

In the third variant, using a laser spot with a large-diameter Gaussian energy distribution, “Wide Gauss” (WG) (Table 1), a spheroid with a very large gel volume was transferred. This mode allowed only solitary spheroids to be transferred. When several spheroids were close together, they were all transferred to the acceptor simultaneously. In addition, the “Wide Gauss” mode resulted in an extensive spattering of the transferred gel drops onto the acceptor surface. Due to the low resolution, this mode was considered unsuitable for the practical implementation of printing with cell spheroids after the first trial experiments.

### 3.2. Features of Laser Transport of Cellular Spheroids Using a Non-Gaussian Intensity Distribution Profile

The search for optimal transfer modes of cell spheroids was carried out using a non-Gaussian intensity distribution profile in variants “Ring and Dot” (RD) and “Two Rings” (TR) (Table 1).

In the RD mode, the main energy of the laser impact falls on the peripheral region of the spheroid (Figure 3). This fluence distribution allows the creation of a gel jet with a characteristic initial transverse size approximately equal to the spheroid diameter. In this case, the presence of the maximum fluence in the central part of the laser spot can lead to undesirable thermal damage to that part of the cell spheroid that is directly adjacent to the paraxial region of the absorbing coating of the donor plate [29]. However, the advantage of this mode is the much smaller transverse pressure gradients acting on the spheroid, which means that the transfer of the spheroid can be gentler compared to the transfer mode with a Gaussian distribution of the laser beam with the same waist radius.

In the TR mode, the energy of the laser action is distributed over two concentric rings, one of which acts on the central part of the spheroid and the second on the peripheral region of the gel at the spheroid boundary (Figure 3). This fluence distribution allows for the creation of a gel jet with a characteristic transverse size slightly larger than the spheroid diameter. In contrast to the RD regime, in which there is a local minimum in the central part of the laser spot; therefore, the probability of thermal damage to the cell spheroid in its paraxial region is significantly reduced compared to the RD mode [28]. Note that due to the larger jet diameter, the TR mode creates smaller transverse pressure gradients compared to the RD mode.

Experiments on the transfer of spheroids in the RD and TR modes showed, as expected, that this process, as in the case of laser beams with a Gaussian energy distribution [21], is determined, in addition to the geometric parameters of the laser waist, by the energy of the laser pulse. If the energy was insufficient, the transfer of the spheroid did not occur, and only a sagging drop was formed in the gel layer, which then returned to the thickness of the gel. At excessive energy values, the formation of multiple jets and spraying of the hydrogel occurred. As a result, spheroids were not gently transferred, as in the case of transfer by single jets, but began to fall apart during the transfer and finally disintegrate into shapeless aggregates upon landing on the surface of the acceptor substrate. In addition, the destruction of the shape of the spheroid can lead to the breaking of the formed bonds between its individual cells, which significantly worsens the quality of emerging biological tissues and the bioprinting process as a whole.

Figure 4 shows successive frames obtained by high-speed shooting during the transfer of cell spheroids. It can be seen that both regimes led to the formation of a jet, while the features of its formation and droplet detachment significantly depended on the laser exposure regime.

The red arrow in Figure 4 shows the position of the spheroid during its transfer. When using the RD (1) mode, a very thin jet of gel is initially formed, visible in the first frame (*t* = 0). Then, a thicker jet of gel is formed, and a spheroid surrounded by the gel appears (*t* = 0.4 ms). The spheroid is subsequently followed by a laminar gel jet (*t* = 1.6–6 ms). In the TR mode (2), no gel jet is observed before the spheroid exit (*t* = 0 and *t* = 0.4 ms). It is followed, as in the first case, by a jet of gel (*t* = 1.6–6 ms). In these modes, there are no ejections of gel microdroplets away from the transfer direction, which indicates optimally selected parameters of the laser action on the spheroid.

Figure 5 shows a schematic position of the recording camera, donor and acceptor plates, and spheroids before and after the transfer as well as optical photographs of the donor and acceptor plates from the side of the donor plate before and after the process of laser transfer of a cell spheroid using the TR mode with a pulse energy of 206 μJ.

Before the transfer, the photograph shows two spheroids (1) and (2) located on the donor plate (Figure 5b). After the transfer of spheroid 1 to the acceptor plate, it is assigned the number (3) (Figure 5a). After the transfer, spheroid (3) is visible to the right of the spheroid (2) in the photograph (Figure 5b) due to the fact that the digital camera was installed at an acute angle to the plate surfaces. After the transfer, a trace of laser impact (4) and bubbles in the gel (5) appears in place of the spheroid (1) on the photograph (Figure 5b).

After exposure to a laser pulse, significant changes appear in the image (Figure 5b): (1) the spheroid (1) disappears from the donor substrate; (2) at this place, a trace of laser exposure (4) and bubbles in the gel (5) appear on the donor plate; (3) next to the image of the spheroid (2) located on the donor plate, the image of the spheroid (3) appears on the acceptor plate.

Note that the traces in the absorbing film (4), depending on the regimes and energies under laser irradiation, had a diameter from 170 to 270 μm. Figure 6 shows an optical photograph of a trace on a donor plate after exposure to a laser pulse in the TR mode with an energy of 206 μJ.

The observed increase in brightness in the regions of the rings when shooting in transmitted light can be associated with a decrease in the thickness of the titanium film in this area. Figure 6b shows the thickness distribution profile of the titanium film calculated from the brightness of the optical image pixels in the plane of the laser spot. It can be seen that, in the region of the rings, the thickness of the Ti film in some places decreases to 10 nm. At the same time, between the rings, as well as in the center of the spot, the film thickness practically did not decrease. 

In more detailed images of the track section in transmitted (Figure 6c) and reflected (Figure 6d) light, it can be seen that microscopic regions of the Ti film partially remained in the region of the rings on the donor plate. At the same time, separate microscopic regions without a film are distinguished in the areas adjacent to the rings. All this suggests that the ablation of the Ti film occurred not due to its evaporation but as a result of detachment from the substrate surface, which can be better from the point of view of reducing the influence of negative factors during laser bioprinting [17]. 

### 3.3. Survival of Cell Spheroids after Laser Transfer

In experiments to test the survival of spheroids after laser bioprinting, three previously selected variants of the intensity distribution in the laser spot were used (see Table 1): “Adapted Gauss”, “Ring and Dot” and “Two Rings”. Data on the survival of cell spheroids after different transfer modes compared with the control are shown in Figure 7. It was expected that the lowest cell survival and the highest percentage of dead cells (56.2% ± 5.8% and 43.8% ± 12.5%, respectively) was for spheroids transferred using the “Adapted Gauss” (AG) mode with the highest fluence value (Figure 7). It should also be noted that, after transfer, the spheroids usually crashed, losing their integrity, while the cells acquire a spherical shape (demonstrating significant damage). In addition, when exposed to such a Gaussian beam, the spheroids often flew away from the desired trajectory.

When using the mode with the “Ring and Dot” (RD) intensity distribution in the laser spot and the laser pulse energy *E* = 95 μJ (fluence *F* = 0.16 J/cm^2^), more living cells and fewer dead cells were observed in spheroids (67.3% ± 14.1% and 32.7% ± 12.3%, respectively) compared to AG group. At the same time, the transfer no longer led to spheroid integrity loss and cell spheroidization. The use of the mode with intensity distribution in a “Two Rings” (TR) laser spot with pulse energy *E* = 206 μJ (fluence *F* = 0.17 J/cm^2^) ensured the highest cell survival and the lowest number of dead cells: (76.6% ± 13.0% and 23.4% ± 6.8%, respectively). 

### 3.4. Advantages and Disadvantages of Different LIFT Modes

The purpose of this work was to demonstrate the advantages of laser printing of such relatively large and, at the same time, delicate biological objects as cellular spheroids by using a spatial distribution of intensity in a laser spot close to Π-shaped. For LIFT of cell spheroids, using the experimental setup (Figure 1), two main printing modes were implemented with a close to Π-shaped intensity distribution: “Ring and Dot” and “Two Rings” (see Table 1). For comparison, we used the standard mode with a Gaussian intensity distribution in the Adapted Gauss beam with a laser spot diameter of *2ω*_0_ = 150 μm, close to the size of a cell spheroid. Figure 8 shows a diagram showing the main advantages and disadvantages of various laser transfer modes.

In the case of using a laser spot with a Gaussian intensity distribution and a small waist radius (TG), a rapidly expanding region with a vapor-gas bubble [13,16] with transverse dimensions smaller than the spheroid diameter is formed in the region above the spheroid top (Figure 8). The resulting high values of the pressure gradient associated with the high irregularity of the distribution of the intensity of the laser impact, in this case, will lead to the destruction of the spheroid. Another negative factor will be high thermal effects on cells in the region of the optical axis due to the high fluence in the center of the beam (8.5 J/cm^2^), which can lead to cell death.

To avoid these two negative factors in bioprinting, we used a wide laser spot with a Gaussian intensity distribution (WG). However, in this case, a gel jet with cross dimensions much larger than the spheroid diameter is formed (Figure 8). Laser printing with such large gel drops leads to a significant decrease in resolution and a decrease in the productivity of the method since printing with single spheroids requires placing them on a donor plate at significantly greater distances from each other.

As a compromise, a laser spot with a Gaussian intensity distribution was used, with a size adapted to the diameter of the cellular spheroid (AG), providing a relatively stable transfer of cellular spheroids but not providing transfer without damage to the cellular spheroid.

Qualitatively better results, as shown in the scheme (Figure 8), can be achieved by bioprinting using a spatial intensity distribution in the laser spot close to Π-shaped (RD and TR). In these regimes, it is possible to obtain the optimal transverse size of the gel jets, which ensures a gentle transfer mode. In this case, the TR mode is preferable since there is a hot spot in the RD distribution in the region of the optical axis, which increases the likelihood of thermal damage to peripheral cells in the upper area of the spheroid [29].

Quantification (Figure 7) confirmed the hypothesis that the intensity distribution in the laser spot influences cell survival. The AG distribution spot contained statistically fewer live cells (56.2% ± 5.8%) than the TR distribution (76.6% ± 13.0%). For RD distributions, the live cell value was intermediate (67.3% ± 14.1%). Quite a high cell viability (~80%) achieved for Two Rings (TR) laser spot geometry is comparable to those of other bioprinting techniques [9,31,32,33,34].

To demonstrate the possibility of performing precision printing with cell spheroids, a line structure was printed (Figure 9) using “Two Rings” intensity distribution in the laser spot. In this case, the process of aiming the laser beam at the spheroid and the movement of the acceptor substrate was controlled manually using two joysticks. The first one made it possible to move the donor plate with spheroids deposited on it, placing the spheroids in the area of laser radiation focusing. The second one was used to displace the acceptor plate to form a line of transferred cell spheroids. The average deviation of the real centers of cell spheroids from the targeting points was 62 ± 33 µm, which is significantly less than the size of the cell spheroid itself. This was enough for demonstration experiments as “proof of concept”, but in the future, this process can be fully automated.

## 4. Conclusions

In this article, we have studied the possibility of gentle and, at the same time, sufficiently accurate laser transfer of cell spheroids. For this, a thorough study of the spheroid transfer and its modes was carried out using various intensity distributions in the laser spot. It has been shown that laser beams with a Gaussian intensity distribution in the laser spot are not optimal for the transport of cell spheroids. A Gaussian beam with “standard” laser spot parameters (spot size is much smaller than the spheroid size), which is used for printing single cells, damages the spheroid and does not give accuracy during the spatial transfer process. The Gaussian beam adapted to the size of the spheroid allows the transfer of large cell aggregates but leads to their partial thermal damage. A wide Gaussian beam with a spot size much larger than the spheroid also transports cellular spheroids but cannot be used due to very poor spatial resolution.

It was shown that laser beams with a non-Gaussian intensity distribution profile in a spot close to a Π-shaped (“Ring and Dot” and “Two Rings”) with a laser spot size comparable to the size of a spheroid make it possible to transfer spheroids in a gentle mode, providing good cell survival. In this case, the “Two Rings” mode is the most preferable since, compared to the “Ring and Dot” mode, it provides smaller gradients of the radial pressure drop and less thermal damage to cells. It has been shown that transfer in selected modes provides survival at the level of control groups.

As a result of this work, the fundamental applicability of the laser-induced forward transfer (LIFT) method for the delicate transfer of spheroids with high viability was proved. The well-known advantages of the LIFT technology, as well as the possibility of its improvement in terms of accelerating and automating the bioprinting process, can make it a leader and make it the gold standard both for bioprinting of various tissue-engineered structures and for the production of various models of biological tissues, for example, of specific spheroid-based tumor-on-a-chip models.

## Figures and Tables

**Figure 1 micromachines-14-01152-f001:**
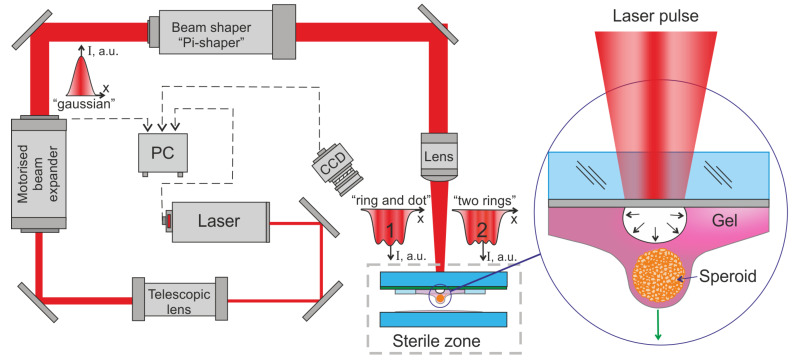
Scheme of experimental setup for laser bioprinting with cell spheroids. On the right, a cell spheroid is shown at the initial moment of its transfer with a bioink jet. Possible forms of intensity distributions in laser beams for the “Ring and Dot” (1) and “Two Rings” (2) modes are shown.

**Figure 2 micromachines-14-01152-f002:**
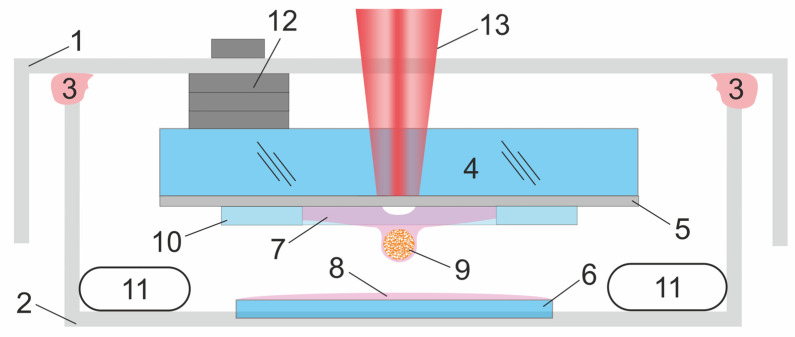
Scheme of the sterile zone with donor and acceptor plates. 1—Petri dish lid, 2—Petri dish bottom, 3—medical vaseline, 4—donor plate, 5—metal absorbing layer, 6—acceptor plate, 7—bioink, 8—acceptor gel, 9—cell spheroid, 10—spacer, 11—wetted cotton swabs, 12—flat magnets, 13—laser beam.

**Figure 3 micromachines-14-01152-f003:**
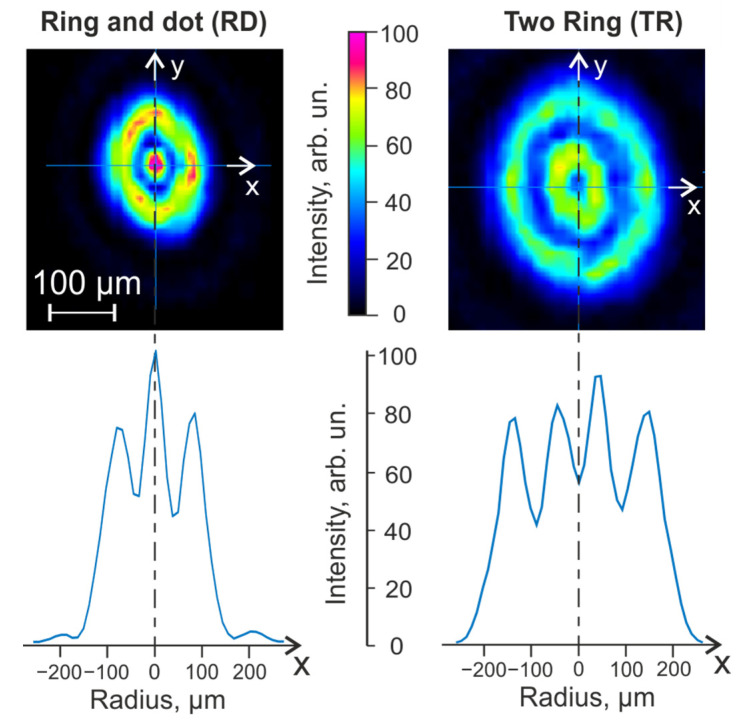
Laser irradiation intensity distribution in the absorbing layer of donor substrate. For “Ring and Dot” mode the energy was *E_RD_* = 95 μJ and for “Two Rings” mode the energy was *E_TR_* = 206 μJ. The corresponding intensity distribution profiles along the X-axis are shown below.

**Figure 4 micromachines-14-01152-f004:**
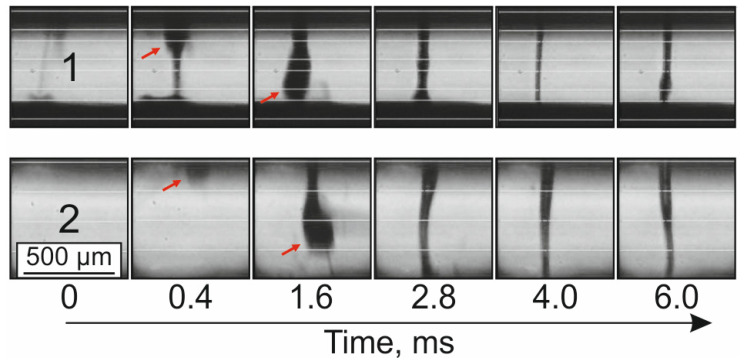
Frames of high-speed shooting during the transfer of cell spheroids with gel jets for the RD (1) and TR (2) laser exposure modes. The red arrow marks the position of the spheroids.

**Figure 5 micromachines-14-01152-f005:**
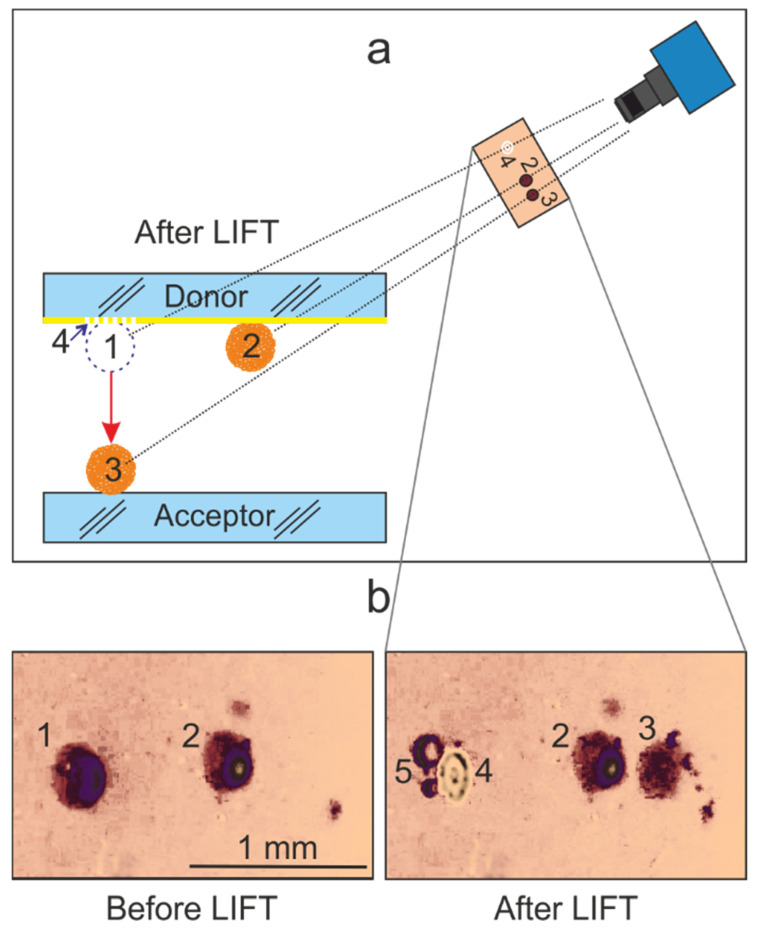
(**a**)—schematic representation of the position of the donor and acceptor plates, as well as the camera that registers the laser transfer of spheroids. Dotted circle 1 shows the location of spheroid 3 on the donor plate before its transfer; 2—another spheroid on the donor plate. (**b**)—optical photographs before LIFT and after LIFT of the cell spheroid. 1—spheroid on a donor plate before transfer; 2—another spheroid on the donor plate; 3—spheroid 1 transferred to the acceptor plate; 4—trace formed in the absorbing film of the donor plate after the transfer of spheroid 1; 5—bubbles in the gel on the donor plate.

**Figure 6 micromachines-14-01152-f006:**
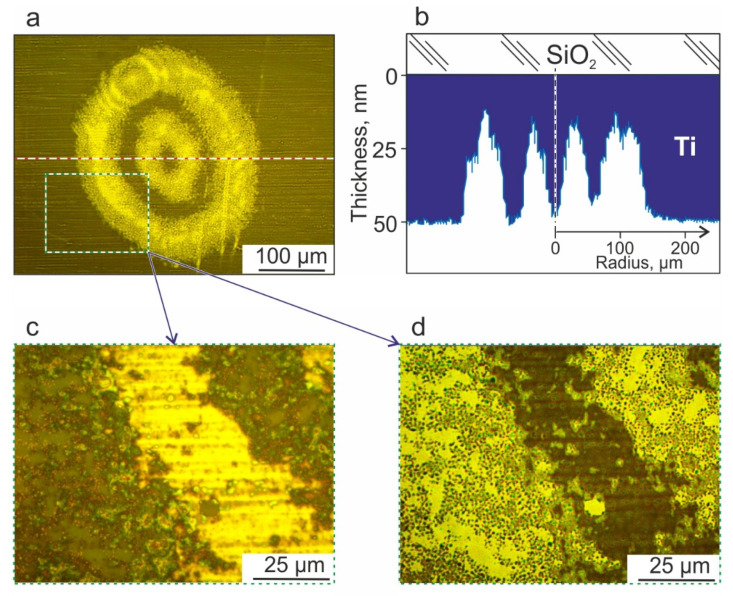
(**a**) Optical photograph of a trace on a donor plate after exposure to a laser pulse with an energy distribution of the TR type. (**b**) Model representation of the titanium film thickness distribution of the donor plate. The data were obtained by analyzing the brightness of the pixels of the optical image along the cross section through the center of the laser spot (red dotted line in (**a**)). Optical photograph of a fragment of the spot area (rectangle in (**a**)) in transmitted light (**c**) and in reflected light (**d**).

**Figure 7 micromachines-14-01152-f007:**
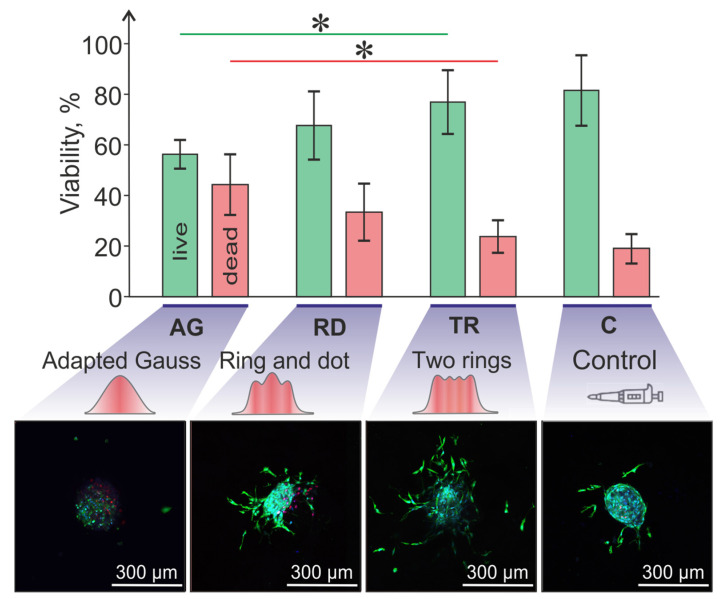
Viability of cell spheroids after LIFT depending on the laser spot shape compared to control group when the spheroids were transferred with a micropipette. The chart shows the percent of live/dead cell in spheroid (* *p* < 0.05). The lower part shows microscopic images of spheroids in test with live/dead assay.

**Figure 8 micromachines-14-01152-f008:**
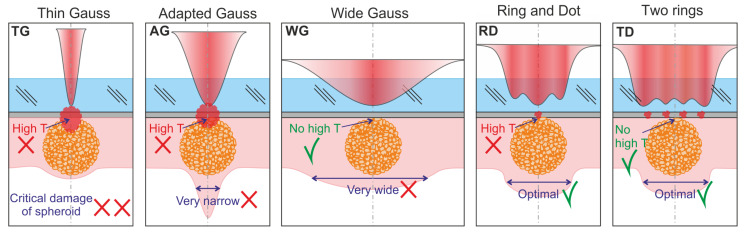
Scheme of the initial stage of cell spheroid transfer for different intensity distributions in the laser spot. The red circles show the regions where the spheroid is affected by high temperatures located near the absorbing film of the donor substrate. The double purple arrow shows the transverse dimensions of the resulting gel jets. Parameters of laser beams with a Gaussian distribution: TG: *E* = 30 μJ, *2ω*_0_ = 30 μm; AG: *E* = 120 μJ, *2ω*_0_ = 150 μm; WG: *E* = 1 μJ, *2ω*_0_ = 1 mm; RD: *E* = 95 μJ, *D_max_* = 240 μm; TR: *E* = 206 μJ, *D_max_* = 360 μm.

**Figure 9 micromachines-14-01152-f009:**
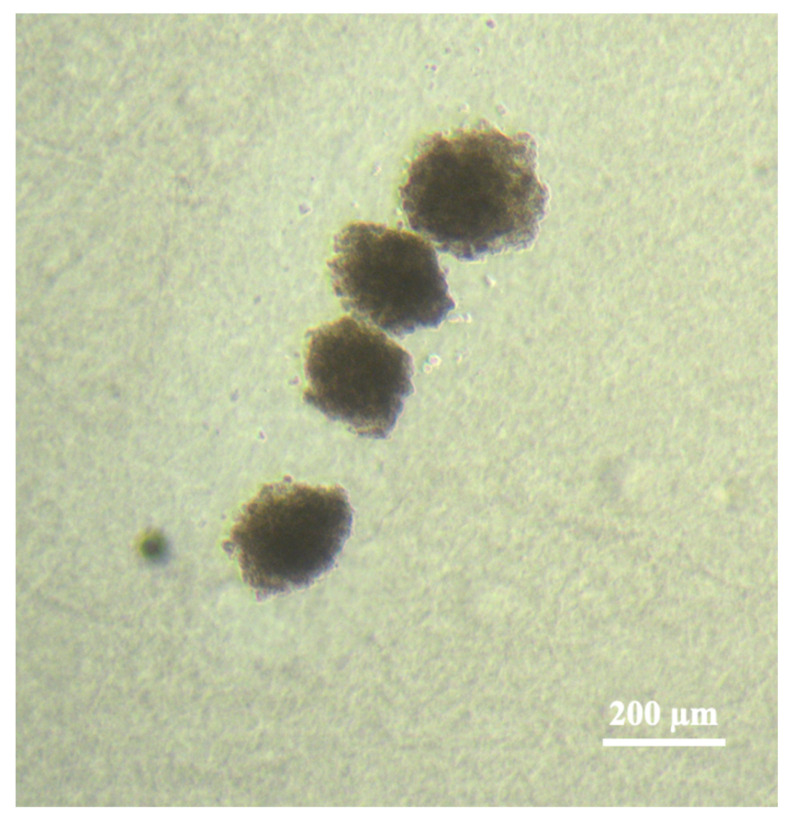
Line structure from four spheroids formed by LIFT using the “Two Rings” intensity distribution in the laser spot.

**Table 1 micromachines-14-01152-t001:** Types of the intensity distribution in the laser spot used for bioprinting with cell spheroids and their phantoms.

Type of Intensity Distribution in the Laser Spot	Conventional Name of the Intensity Distribution	Laser Spot Diameter, μm	Laser Pulse Energy, μJ
Gaussian	Thin Gauss	*D*_TG_ = 30	*E*_TG_= 80 ± 5
Gaussian	Adapted Gauss	*D*_AG_ = 150	*E*_AG_ = 120 ± 15
Gaussian	Wide Gauss	*D_WG_*~1000	*E_WG_*~1000
non-Gaussian	Ring and Dot	*D_RD_* = 200	*E_RD_* = 95 ± 6
non-Gaussian	Two Rings	*D_TR_* = 350	*E_TR_* = 206 ± 10

## Data Availability

The data presented in this study are available on request from the corresponding author.

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
