# Peer review of "Laser Bioprinting with Cell Spheroids: Accurate and Gentle"

_micromachines, 2023, doi:10.3390/mi14061152_

Round 1
Reviewer 1 Report
This paper proposed an approach to printing gently with cell spheroids by laser-induced direct transfer. This method is realized by adding a new optical part based on the Pi-Shaper element into the laser bioprinter, which allows for forming laser spots with different non-Gaussian intensity distributions. It ensures good cell survival and high resolution of laser bioprinting. In general, the manuscript is well written. However, the following questions should be addressed.
1. In the paper, the author mentioned that this method can achieve high-resolution laser bioprinting. Please use experiments/data to prove the resolution that can be achieved.
2. Can the width of the laser beam be regulated? If so, what effect does width have on printing?
3. Two headings “3.1” appear in the paper, and the second heading “3.1” is the same as the heading “3.1”.
This paper proposed an approach to printing gently with cell spheroids by laser-induced direct transfer. This method is realized by adding a new optical part based on the Pi-Shaper element into the laser bioprinter, which allows for forming laser spots with different non-Gaussian intensity distributions. It ensures good cell survival and high resolution of laser bioprinting. In general, the manuscript is well written. However, the following questions should be addressed.
1. In the paper, the author mentioned that this method can achieve high-resolution laser bioprinting. Please use experiments/data to prove the resolution that can be achieved.
2. Can the width of the laser beam be regulated? If so, what effect does width have on printing?
3. Two headings “3.1” appear in the paper, and the second heading “3.1” is the same as the heading “3.1”.
Author Response
Reviewer #1.
Dear reviewer. We thank you for your comments, and we provide answers to them.
- In the paper, the author mentioned that this method can achieve high-resolution laser bioprinting. Please use experiments/data to prove the resolution that can be achieved.
Thank you for your comment.
To prove the resolution of this method of laser bioprinting, experiments to printing various geometrical structures, including lines (see Fig. 9) were carried out. The average deviation of the real centers of cell spheroids from the targeting points was 62 ± 33 µm, which is significantly less than the size of the cell spheroid itself. This resolution was obtained at the distance between the donor and acceptor plates ~0.8 mm with optimal regime with the “Two Rings” intensity distribution in the laser spot. We believe that such a deviation is primarily due to the imperfection of the shape of the spheroids, and the deviation of the spheroid center from the optical axis of the beam due to manually aiming the laser beam at the spheroid and the movement of the acceptor substrate using two joysticks.
The relevant text has been added to the article.
Abstract section:
Text added: “The proposed method showed a high spatial resolution of laser printing of cell spheroid geometric structures at the level of 62 ± 33 µm, which is significantly less than the size of the cell spheroid itself.”
Last paragraph of Section “3. Results and Discussion”, before section “Conclusion”:
Text added: “The average deviation of the real centers of cell spheroids from the targeting points was 62 ± 33 µm, which is significantly less than the size of the cell spheroid itself.”
- Can the width of the laser beam be regulated? If so, what effect does width have on printing?
Yes, changing width of the laser beam can effect on bioprinting process.
If the width of the laser beam is the width (size, diameter) of the laser spot, then it can be adjusted by changing the width or divergence of the laser beam before entering the P-shaper optical element, as well as by moving the working plane (absorbing layer on the donor plate) along the optical axis.
The size of the laser spot significantly affects the bioprinting process (see Fig. 8 and its commentary). The experiments showed that the optimal size of the laser spot should approximately correspond to the size of the spheroid. As the spot size increases, the resolving power deteriorates. Reducing the spot size at the same laser pulse energy leads to greater damage to the cell spheroid.
We checked the text of the article and corrected it to better describe the role of the laser beam width and its divergence.
Section “2. Materials and Methods”, 3 paragraph after Figure 1:
Text added\corrected: It is necessary to set the spot size and the shape of the intensity distribution in the laser spot to adjust the system for the required laser bioprinting mode. For this, one should sequentially adjust the laser beam diameter at the entrance to the Pi-Shaper and the divergence of the laser irradiation, as well as the distance of the working plane (absorbing layer on the donor plate) from the focal point.
- Two headings “3.1” appear in the paper, and the second heading “3.1” is the same as the heading “3.1”.
Thank you for your consideration in reviewing our article. We have corrected this shortcoming.
Reviewer 2 Report
Manuscript ID - micromachines-2361370
Title - Laser bioprinting with cell spheroids: accurate and gentle
Authors - Ekaterina Dmitrievna Minaeva et al.
Suggestion: Major revision
The manuscript demonstrates bioprinting with cell spheroids by laser-induced direct transfer. Authors claim to ensure good cell survival. Although the manuscript is well written, there are some concerns that should be addressed before further consideration:
1. Authors should include few of the results and conclusion in the abstract.
2. How did authors set different specific parameters like wave length?
3. Did authors study different micro jet evolutions and drop formation at different laser energies?
4. What was the effect of various laser intensities on the number of cells per droplet?
5. Authors should discuss about the severe genomic instabilities caused by lasers during bioprinting especially at the wavelength used in the manuscript.
There are a few grammatical errors in the manuscript. Authors need to check the whole manuscript for the grammatical errors.
Author Response
Reviewer #2.
Dear reviewer. We thank you for your comments, and we provide answers to them.
- Authors should include few of the results and conclusion in the abstract.
Thank you for your comment. We have improved the text by explicitly highlighting a few of the results and conclusion in the abstract.
The relevant text has been added to the article.
Abstract section:
Text added\corrected:
“The possibilities of cell spheroids printing by laser-induced forward transfer in a gentle mode, which ensures good cell survival ~80% without damage and burns, were demonstrates.”
“The proposed method showed a high spatial resolution of laser printing of cell spheroid geometric structures at the level of 62 ± 33 µm, which is significantly less than the size of the cell spheroid itself.”
“It is shown that laser spots with an intensity distribution profile of the "Two rings" type (close to Π-shaped) and a size comparable to a spheroid are optimal.”
- How did authors set different specific parameters like wave length?
Thanks for your question. The wavelength of laser radiation is extremely important for the effect of direct radiation on cells. The negative impact on average decreases with increasing wavelength from ultraviolet to near infrared due to a decrease in photon energy. Therefore, the impact in the near-IR region is more gentle in this sense. In our case, the laser radiation is mainly absorbed in the metal layer on the donor substrate and only about 10% of the incident intensity irradiates the cells [1,2]. Our scientific group attaches great importance to minimizing the negative effect on cells during printing, including through direct laser irradiation. In this regard, laser radiation in the near-IR range was chosen.
- Grosfeld, E. V.; Zhigarkov, V.S.; Alexandrov, A.I.; Minaev, N. V.; Yusupov, V.I. Theoretical and Experimental Assay of Shock Experienced by Yeast Cells during Laser Bioprinting. Int. J. Mol. Sci. 2022, 23, doi:10.3390/ijms23179823.
- Yusupov, V.I.; Zhigar’kov, V.S.; Churbanova, E.S.; Chutko, E.A.; Evlashin, S.A.; Gorlenko, M. V; Cheptsov, V.S.; Minaev, N. V; Bagratashvili, V.N. Laser-Induced Transfer of Gel Microdroplets for Cell Printing. Quantum Electron. 2017, 47, 1158–1165, doi:10.1070/QEL16512.
- Did authors study different micro jet evolutions and drop formation at different laser energies?
Our scientific group pays great attention to the study of these processes. We have studied in detail different micro jet evolutions and drop formation using high-speed video recording and shadow photography [1,2,3,4]. This information allowed us to select the optimal parameters for laser bioprinting with microorganism cells.
As part of this work, we also used high-speed video recording to search for optimal modes of transfer of large objects - cellular spheroids (Fig. 4). At the same time, studies were carried out both with cellular spheroids and with their phantoms in various variants of the distributions of intensity in the laser spot and energy in the laser pulse. We were able to preliminarily determine the boundaries of optimal regimes by analyzing the shape of jets and drops by high-speed video. However, constant high-speed video recording during the “working modes” of bioprinting was very difficult due to the sterile zone, which we installed in the laser bioprinter. So we used an overview camera (Fig. 5), which was used to aim and control the result of bioprinting.
- Yusupov, V.I.; Zhigar’kov, V.S.; Churbanova, E.S.; Chutko, E.A.; Evlashin, S.A.; Gorlenko, M. V; Cheptsov, V.S.; Minaev, N. V; Bagratashvili, V.N. Laser-Induced Transfer of Gel Microdroplets for Cell Printing. Quantum Electron. 2017, 47, 1158–1165, doi:10.1070/QEL16512.
- Mareev, E.; Minaev, N.; Zhigarkov, V.; Yusupov, V. Evolution of Shock-Induced Pressure in Laser Bioprinting. Photonics 2021, 8, 374, doi:10.3390/photonics8090374.
- Yusupov, V.; Churbanov, S.; Churbanova, E.; Bardakova, K.; Antoshin, A.; Evlashin, S.; Timashev, P.; Minaev, N. Laser-Induced Forward Transfer Hydrogel Printing: A Defined Route for Highly Controlled Process. Int. J. Bioprinting 2020, 6, 1–16, doi:10.18063/ijb.v6i3.271.
- Cheptsov, V.; Minaev, N.; Zhigarkov, V.; Tsypina, S.; Krasilnikov, M.; Gulyashko, A.; Larionov, I.; Tyrtyshnyy, V.; Gonchukov, S.; Yusupov, V. Laser Bioprinting without Donor Plate. Laser Phys. Lett. 2022, 19, 085602, doi:10.1088/1612-202X/ac7b32.
- What was the effect of various laser intensities on the number of cells per droplet?
The purpose of our work is to determine the transfer regime of a single cell spheroid in one hydrogel drop. In our case, the distribution of intensity in the laser spot and size of the spot must correspond with the diameter of cell spheroid (which is proportional to the number of cells). Therefore, in general, the intensity primarily determines the mode of printing the cellular spheroid. If the in laser spot intensity is insufficient, the cell spheroid will not be transferred. At high intensity, the printing process will be imprecise, and the cell spheroid itself may be damaged due to the strong pressure gradient and the transmitted laser radiation.
- Authors should discuss about the severe genomic instabilities caused by lasers during bioprinting especially at the wavelength used in the manuscript.
Thank you for your attention to this aspect.
To reduce the effect on cells, we chose the wavelength of the near-IR range. In addition, the laser radiation is mainly absorbed in the metal layer on the donor substrate and only about 10% of the incident intensity irradiates the cells [1,2]. As for genomic instability, the most destructive effect in this wavelength range can only be exerted by ultrashort pulses, which provide multiphoton absorption [3]. The studies carried out in [4] showed that even the use of shorter wavelength ultraviolet radiation of 355 nm with a pulse duration of 10 ns (approximately as in our work) leads to negligible genotoxic stress during laser bioprinting.
- Grosfeld, E. V.; Zhigarkov, V.S.; Alexandrov, A.I.; Minaev, N. V.; Yusupov, V.I. Theoretical and Experimental Assay of Shock Experienced by Yeast Cells during Laser Bioprinting. Int. J. Mol. Sci. 2022, 23, doi:10.3390/ijms23179823.
- Yusupov, V.I.; Zhigar’kov, V.S.; Churbanova, E.S.; Chutko, E.A.; Evlashin, S.A.; Gorlenko, M. V; Cheptsov, V.S.; Minaev, N. V; Bagratashvili, V.N. Laser-Induced Transfer of Gel Microdroplets for Cell Printing. Quantum Electron. 2017, 47, 1158–1165, doi:10.1070/QEL16512.
- Botchway, S.W.; Reynolds, P.; Parker, A.W.; O’Neill, P. Use of near Infrared Femtosecond Lasers as Sub-Micron Radiation Microbeam for Cell DNA Damage and Repair Studies. Mutat. Res. Mutat. Res. 2010, 704, 38–44, doi:10.1016/j.mrrev.2010.01.003.
- Karakaidos, P.; Kryou, C.; Simigdala, N.; Klinakis, A.; Zergioti, I. Laser Bioprinting of Cells Using UV and Visible Wavelengths: A Comparative DNA Damage Study. Bioengineering 2022, 9, 378, doi:10.3390/bioengineering9080378.
Round 2
Reviewer 2 Report
Authors have done significant changes in the revised manuscript and hence, it can be accepted in the current form.
Author Response
Dear Reviewer,
We thank you for your work on improving the quality of our article.